# Exploring Plastomic Resources in *Sempervivum* (Crassulaceae): Implications for Phylogenetics

**DOI:** 10.3390/genes15040441

**Published:** 2024-03-30

**Authors:** Junhu Kan, Shuo Zhang, Zhiqiang Wu, De Bi

**Affiliations:** 1Shenzhen Branch, Guangdong Laboratory of Lingnan Modern Agriculture, Key Laboratory of Synthetic Biology, Laboratory of the Ministry of Agriculture and Rural Affairs, Agricultural Genomics Institute at Shenzhen, Chinese Academy of Agricultural Sciences, Shenzhen 518120, China; junhu.kan21@alumni.xjtlu.edu.cn (J.K.); szhang@webmail.hzau.edu.cn (S.Z.); 2College of Landscape Engineering, Suzhou Polytechnic Institute of Agriculture, Suzhou 215000, China

**Keywords:** Crassulaceae, *Sempervivum*, plastome, codon usage bias, hypervariable regions, phylogenetic analysis

## Abstract

The plastid organelle is vital for photosynthesis and energy production. Advances in sequencing technology have enabled the exploration of plastomic resources, offering insights into plant evolution, diversity, and conservation. As an important group of horticultural ornamentals in the Crassulaceae family, *Sempervivum* plants are known for their unique rosette-like structures and reproduction through offsets. Despite their popularity, the classification status of *Sempervivum* remains uncertain, with only a single plastome sequence currently available. Furthermore, codon usage bias (CUB) is a widespread phenomenon of the unbalanced usage of synonymous codons in the coding sequence (CDS). However, due to the limited available plastid data, there has been no research that focused on the CUB analysis among *Sempervivum* until now. To address these gaps, we sequenced and released the plastomes of seven species and one subspecies from *Sempervivum*, revealing several consistent patterns. These included a shared 110 bp extension of the rps19 gene, 14 hypervariable regions (HVRs) with distinct nucleotide diversity (π: 0.01173 to 0.02702), and evidence of selective pressures shaping codon usage. Notably, phylogenetic analysis robustly divided the monophyletic clade into two sections: Jovibarba and Sempervivum. In conclusion, this comprehensive plastomic resource provides valuable insights into *Sempervivum* evolution and offers potential molecular markers for DNA barcoding.

## 1. Introduction

The Crassulaceae family, the largest family in Saxifragales, contains around 1400 species across three subfamilies: Sempervivoideae, Kalanchoideae, and Crassuloideae. The Sempervivoideae subfamily comprises five major clades (Telephium, Aeonium, Acre, Leucosedum, Sempervivum) [1,2,3,4,5,6]. And many species in this family were seen as ornamental with highly economic values [7,8,9]. Notably, the genus *Sempervivum*, also known as hen and chicks or houseleeks, is the sole member of the Sempervivum clade. Native to the mountains of Europe, Africa, and Asia, *Sempervivum* plants are characterized by their rosette-like structures composed of succulent leaves and their ability to reproduce via offsets. There are over 40 species within this genus, exhibiting a wide range of colors, shapes, and sizes [10]. However, the classification status of *Sempervivum* remains uncertain, with some studies suggesting that it should be divided into two separate genera, namely, *Jovibarba* and *Sempervivum*. This proposal is based on the morphological and genetic differences found between these two groups [10,11,12].

In plant cells, plastids are crucial organelles responsible for photosynthesis and energy production [13,14,15]. The plastome usually possesses a conservative quadripartite circular structure with two single-copy regions (LSC and SSC), separated by two inverted repeat (IR) regions [16,17,18,19]. Plastomes are valuable tools for inferring phylogenetic relationships between plants due to their unique characteristics [20,21,22,23]. With the rapid development in advanced high-throughput sequencing technologies, thousands of plastomes have been released in public databases. However, to date, only one plastome from *Sempervivum* is available on the NCBI database (GenBank accession number: NC_053954). The insufficiency of plastid data has hindered the progress in exploring the evolutionary status of *Sempervivum*. Therefore, more plastomic resources from this genus could effectively address this issue.

Moreover, codon usage bias (CUB) is a widespread phenomenon of the unbalanced usage of synonymous codons in the coding sequence (CDS). Some codons are used more frequently than their synonymous codons [24,25,26]. There are various indices that are frequently employed to evaluate the CUB, such as the effective number of codons (ENC), relative synonymous codon usage (RSCU), parity rule 2 (PR2) plot, and neutrality plot [27,28]. CUB analysis can serve as a potent technique to uncover evolutionary patterns within taxa or specific genes [16]. A recent study [16] analyzed the CUB among 86 Crassulaceae species and found that the matK gene had a unique codon usage pattern among Crassulaceae with relatively high ENC values. However, due to the limited available plastid data of *Sempervivum*, there has been no research that focused on the CUB analysis among *Sempervivum* until now. Thus, to gain a deeper insight into the phylogeny of *Sempervivum*, it is crucial to explore the synonymous codon usage patterns in *Sempervivum* with a large sampling scale.

In this study, we sequenced and released the plastomes of seven species and one subspecies (*S*. *globiferum*, *S*. *globiferum* subsp. *hirtum*, *S*. *heuffelii*, *S*. *calcareum*, *S*. *funckii*, *S. ciliosum*, *S. arachnoideum*, and *S. tectorum*). Notably, three of these belong to the section *Jovibarba*. By analyzing these new plastomes, we investigated (1) the structural diversity among *Sempervivum* plastomes, (2) the codon usage pattern of protein-coding genes, and (3) the phylogenetic relationships within the genus. Our findings will undoubtedly provide a deeper understanding of the evolution of *Sempervivum* while simultaneously identifying potential molecular markers for DNA barcoding.

## 2. Materials and Methods

### 2.1. Sample Collection, DNA Extraction, and Sequencing

This study collected fresh leaves of eight houseleek (*Sempervivum*) taxa from the greenhouse at the Agricultural Genomics Institute, Chinese Academy of Agricultural Sciences. The samples were assigned the voucher codes KL01699, KL01791, KL01265, KL01614, KL01838, KL02009, KL01220, and KL01460, which represented *S*. *globiferum*, *S*. *globiferum* subsp. *hirtum*, *S*. *heuffelii*, *S*. *calcareum*, *S*. *funckii*, *S. ciliosum*, *S. arachnoideum*, and *S. tectorum*, respectively. The CTAB method [29] was used to extract the total genomic DNA from the plant samples. The sequencing process was finished by Biomarker Technologies (Beijing, China). Generally speaking, library construction was then performed using an Illumina TruSeq DNA PCR-Free Library Prep Kit. Finally, Illumina Hiseq X Ten sequencing technology was utilized to sequence the prepared libraries.

### 2.2. Plastome Assembly and Genome Annotation

The plastomes of the eight taxa were assembled using GetOrganelle v.1.6.4 [30], with the plastomic sequence of *S. tectorum* (NC_053954) as a reference. Subsequently, the assembled plastomes were annotated using the online program GeSeq [31] and CPGAVAS2 [32]. The protein-coding genes (PCGs) were manually checked by comparing with the reference (NC_053954) through a BlastN search [33]. Finally, Chloroplot [17] was utilized to generate circular map visualizations of the plastomes.

### 2.3. Composition and Structural Analysis of Plastomes

The nucleotide composition of the plastomes from eight *Sempervivum* taxa was analyzed using Bioedit [34]. Next, the plastomes were aligned using MAFFT software (v 7.487) [35], and genomic visualization was performed with mVISTA [36] using the Shuffle-LAGAN mode. Additionally, DNA polymorphisms were evaluated with sliding window nucleotide diversity (π) values by utilizing DnaSP v6.12 [37], with a window length of 600 bp and a step size of 200 bp. A high variability region (HVR) was defined as a contiguous sliding window with a π value greater than the average plus 2 standard deviations. The variation in the structures of the SC and IR borders among the eight *Sempervivum* taxa was visualized using IRScope [38].

### 2.4. Analysis of Codon Usage Pattern

To elucidate the codon usage patterns in the PCGs of the eight selected taxa, we employed CodonW v.1.4.2 (http://codonw.sourceforge.net/ (accessed on 28 March 2024)) to calculate several crucial metrics, including the effective number of codons (ENC), relative synonymous codon usage (RSCU), and the G + C content at the third position of synonymous codons (GC3s) across the entire gene sequences. The genes for analysis were selected from the PCGs according to the following criteria: (1) the sequence had a length greater than or equal to 300 bp and was divisible by three, (2) it was initiated with a valid start codon, (3) it was terminated with a recognized termination codon, and (4) it did not contain any intermediate stop codons. By applying these stringent criteria, we ensured the robustness and accuracy of our analysis of codon usage patterns.

The ENC value serves as an indicator of the extent to which synonymous codons are unequally utilized and spans a range from 20 (indication extreme bias) to 61 (suggesting no bias) [39]. On the other hand, the RSCU value signifies the ratio of an observed synonymous codon frequency to its theoretical expected frequency. Consequently, an RSCU > 1 indicates that this codon is employed more frequently than its synonymous counterparts, while an RSCU < 1 designates it as a less frequently used codon [40].

Optimal codons were identified using the ΔRSCU method [41,42]. To achieve this, the PCGs with the top 5% highest and lowest ENC values were selected as the high ENC and low ENC groups, respectively. Then, codons with ΔRSCU values greater than 0.08 and RSCU values meeting the conditions of low ENC group > 1 and high ENC group < 1 were considered optimal codons.

### 2.5. Phylogenetic Analysis for Sempervivum Species

To determine the evolutionary position of the genus *Sempervivum* within the Crassulaceae family, we incorporated eight newly generated datasets into our analysis, along with plastomes from 86 other Crassulaceae species (Appendix A) downloaded from the NCBI database (https://www.ncbi.nlm.nih.gov/). Previous research by Han et al. [43] revealed a sister relationship between Crassulaceae and Haloragaceae. Thus, two species (*Myriophyllum* aquaticum and *M.* spicatum) from Haloragaceae were employed as outgroups. Phylogenetic trees reconstruction was performed by using both maximum likelihood (ML) and Bayesian inference (BI) methods. Our dataset consisted of 79 protein-coding genes (PCGs) from 96 species (Data S1), and we employed RAxML 8.2.12 [44] under the GTRCAT model with 1000 bootstrap replicates for ML tree reconstruction. Convergence was assessed using the “-I autoMRE” parameter. For the BI analysis, we utilized MrBayes 3.2.7a [45] and determined the best models for nucleotide substitutions with ModelTest-NG [46] based on Bayesian information criterion (BIC). Two trials with four independent Markov chains were run for 10 million generations, and convergence was evaluated using Tracer 1.7.1 [47].

## 3. Results

### 3.1. Plastome Organizations and Structural Features

In this study, the plastome organizations and structural features of eight newly reported *Sempervivum* species were investigated. The plastomes ranged in length from 150,845 to 151,473 bp and exhibited a typical quadripartite and circular structure (Figure 1), comprising IR (25,778–25,829 bp), SSC (16,688–16,762 bp), and LSC (82,499–83,151 bp) regions (Appendix A). Notably, all *Sempervivum* plastomes analyzed in this study displayed a similar GC content. The overall GC contents of these plastomes ranged from 37.57% to 37.63%. Interestingly, upon further examination, we found that the GC contents within the IR regions were higher (42.94–42.99%) than those observed in the LSC (35.44–35.51%) and SSC (31.44–31.68%) regions (Appendix A).

Furthermore, a total of 133 genes were identified within the plastomes of the investigated *Sempervivum* taxa, comprising 85 protein-coding genes (PCGs), 36 tRNA genes, 8 rRNA genes, and 4 pseudogenes. The overall organization of the plastomes was analyzed using mVISTA, revealing that the SC regions exhibited higher divergence than the IR regions (Figure 2). Furthermore, when compared with the conserved non-coding regions (CNS), the PCGs displayed a lower level of variation. Sliding-window-based π values were calculated for these new plastomes, ranging from 0 to 0.02702, with an average of 0.003754 and a standard deviation of 0.003955. The high-variability regions were identified by π values that were greater than 0.01166, and a total of 14 HVRs (π value: 0.01173–0.02702) were found in *Sempervivum* (Figure 3, Appendix A). These HVRs can potentially be used as barcodes for identifying species within *Sempervivum*.

This study revealed consistent patterns in the eight newly sequenced *Sempervivum* plastomes (Figure 4). Notably, the IR junction pattern showed remarkable similarity. Specifically, the *ycf1* and *rps19* genes spanned both the SSC and IRa regions, with the majority of them located in the SSC region. Due to their position in the inverted repeat region, fragments of these genes were also present in the IRb region. Furthermore, the *rps19* gene extended 110 bp into the IRb region in all of the newly assembled *Sempervivum* plastomes. Interestingly, this shared structural feature was also observed in other Crassulaceae species [48].

### 3.2. Codon Usage Bias Indices of Plastid Genes from Sempervivum

To determine the rules governing codon usage patterns in *Sempervivum*, we analyzed four codon usage bias indices, namely, RSCU, ENC, GC content, and optimal codons, in the eight newly sequenced plastomes. Our analysis included 53 CDSs with lengths of at least 300 bp. Of these, 59 codons (excluding Met, Trp, and stop codons) were analyzed in the RSCU analysis.

The overall RSCU values in *Sempervivum* ranged from 0.33 (AGC) to 2 (TTA), with a heatmap showing a preference for A/T-ending codons over G/C-ending codons in plastid PCGs (Appendix A, Figure 5). Among these CDSs, the *clpP* gene in *S. heuffelii* displayed the highest ENC value of 54.19, indicating a relative average codon usage in this gene. In contrast, the *rps8* gene in *S. tectorum*, *S. ciliosum*, *S. funckii*, *S. calcareum*, and *S. arachnoideum* had the lowest ENC value of 35.44, indicating an extreme bias in codon usage (Appendix A).

We plotted the ENC and GC3s data from the analyzed plastomes in *Sempervivum* and found that most ENC values were lower than expected, lying below the standard curve (Figure 6). This suggested that the codon usage preferences in *Sempervivum* were influenced by natural selection.

We used a ΔRSCU method to determine the optimal codons (Table 1) in *Sempervivum*, which revealed that all species shared two optimal codons, namely, CUA and CGU. Additionally, *S. tectorum* and *S. calcareum* had an extra codon, namely, GGG. These two species were grouped in a clade in the neighbor-joining tree for *Sempervivum*, implicating that codon usage contains a phylogenetic signal.

After analyzing the neutrality graph, we examined the relationship between GC12 (average GC content at the first and second positions of codons) and GC3 (Appendix A). This investigation aimed to identify the predominant influencing factor, either natural selection or mutation pressure, shaping the codon usage patterns. The GC12 values ranged from 0.3100 to 0.5652, while the GC3 values spanned from 0.1875 to 0.3669. A strong GC12-GC3 correlation implies mutation-based CUB, while a weak or absent correlation suggests that translation selection is the driving force. In *Sempervivum*, the regression coefficient, which signifies the slope of the neutrality plot, between GC12 on GC3 exhibited a range from 0.1953 to 0.2087, with an average value of 0.2039. This observation implies that the compositional characteristics of the first two bases within codons in *Sempervivum* may differ from those of the third base. Furthermore, the results show a relative neutrality and relative constraint of 20.39% and 79.61% for GC3, respectively. The GC12 in *Sempervivum* was influenced by mutation pressure and natural selection, with a ratio of 0.2039/0.7961 = 0.2561. These findings demonstrate that natural selection primarily impacted the codon usage patterns within the coding sequences of plastomes across the *Sempervivum*.

### 3.3. Phylogenetic Analysis of Sempervivum

To shed light on the phylogenetic relationships between *Sempervivum* species within the Crassulaceae family, we employed 86 additional plastome sequences obtained from the Crassulaceae species and used both ML and BI methods. In addition, we selected two *Myriophyllum* species to serve as outgroups for our analysis (Figure 7).

The resulting phylogeny revealed seven well-supported clades: *Leucosedum*, *Acre*, *Aeonium*, *Sempervivum*, *Telephium*, *Kalanchoe*, and *Crassula*. Among these clades, the first five belonged to the subfamily Sempervivoideae, while the latter two clades represented the subfamilies Kalanchoideae and Crassuloideae, respectively. Eleven species of *Crassula* formed a strongly supported group, known as the Crassula clade, which was the sister group to all other members of the family Crassulaceae. Interestingly, the Kalanchoideae was closely related to the Sempervivoideae, suggesting a recent common ancestry between the two groups. Within the Sempervivoideae subfamily, the Telephium clade formed a sister group to the Aeonium, Leucosedum, Acre, and Sempervivum clades. This clade, which consisted of 45 species across eight genera, could be further divided into two subclades, namely, Hylotelephium and Rhodiola, with high support ([BS] = 100, [PP] = 1.0).

In our study, the eight newly sequenced *Sempervivum* plastomes formed a well-supported monophyletic clade (Sempervivum), which was a sister to the Aeonium, Leucosedum, and Acre clades ([BS] = 100, [PP] = 1.0). Further analysis revealed that the monophyletic clade could be further divided into two sections: Jovibarba and Sempervivum. The Jovibarba subclade contains *S. globiferum* and *S. globiferum* subsp. *hirtum*, which formed a sister pair, and *S. heuffelii*, which was their sister species ([BS] = 100 and [PP] = 1.00). The Sempervivum subclade comprised five species, including *S. tectorum* and *S. calcareum*, which formed a sister pair ([BS] = 95 and [PP] = 1.00), and *S. ciliosum*, which was grouped with them. Additionally, *S. funckii* and *S. arachnoideum* constituted another sister group. The two groups were clearly distinct with robust support ([BS] = 100 and [PP] = 1.00). Notably, within the Sempervivum subclade, all nodes exhibited robust support, with bootstraps scores of at least 95 and posterior probabilities of 1.00.

## 4. Discussion

In this paper, we present the complete plastome sequences of eight *Sempervivum* taxa and conducted a comprehensive analysis of their structure comparison, codon usage and aversion patterns, and phylogenetic analysis. While it is widely recognized that IR regions of plastomes exhibit greater conservation compared with other regions, they are not exempt from evolutionary events, such as contraction and expansion at their borders. These events have been observed frequently in various plastomes [49,50,51,52,53,54,55], highlighting the dynamic nature of plastome evolution.

Our investigation revealed a consistent structural feature at the IR/SC boundary for *Sempervivum*. Specifically, all the plastomes newly reported in the present study had a 110-base pair extension of the *rps19* gene in the IRb region. This 110 bp span was also found in other Crassulaceae species, such as *Crassula* [16], *Aeonium* and *Monanthes* [48], and *sedum* [16]. These observations led to the hypothesis that this 110-base pair extension may serve as a special family marker for Crassulaceae. Our study provides additional compelling evidence to support this proposition.

Moreover, it is crucial to emphasize that prior research revealed two significant phenomena: the loss of the *rps19* gene and the presence of pseudogenes within the IR regions. These occurrences directly impact the length of IRs [56]. Additionally, the dynamic character of plastome evolution is further illustrated by the variations in IR length observed between distinct species and genera, which can be attributed to various factors, including gene loss, gene duplication, and differences in intron–exon structure [57]. These findings underscore the pivotal role of structural changes in shaping these genomes.

The CUB is a crucial aspect of molecular evolution, as it can provide insights into the evolutionary history of species and genera. Previous studies showed that the CUB is species and gene specific [25], and thus, analyzing the CUB can uncover significant phylogenetic relationships between different species. This study aimed to investigate the CUB of 53 CDS from eight newly sequenced plastomes of *Sempervivum* species to gain a deeper understanding of their evolutionary history.

This study employed the ENC plot and neutrality graph to examine the influence of natural selection and mutation pressure on the CUBs of the eight plastomes. The results indicate that natural selection had a stronger influence than mutation pressure in shaping the CUB, suggesting that the codon usage patterns in the plastomes were primarily driven by selective pressures.

Furthermore, this study found that the optimal codon in the plastomes of two species, *S. tectorum* and *S. calcareum*, were closely related to phylogenetic patterns. These two species shared three optimal codons and they were clustered into a clade, while the other six species only shared two optimal codons. This suggests that in the same genus, the species that share the same optimal codons may have a closer evolutionary relationship.

This study contributes to the current understanding of the CUB and its relationship to phylogeny. The findings support the idea that the CUB can serve as a valuable tool for inferring evolutionary relationships between species. Moreover, this study highlights the importance of investigating the CUB in different species and genes to gain a comprehensive understanding of molecular evolution.

Our phylogenetic analysis results presented in this paper provide new insights into the evolutionary relationships within the genus *Sempervivum* and the Crassulaceae family. The resulting phylogeny revealed seven well-supported clades: Leucosedum, Acre, Aeonium, Sempervivum, Telephium, Kalanchoe, and Crassula. The first five clades belonged to the subfamily Sempervivoideae, while Kalanchoe and Crassula represented Kalanchoideae and Crassuloideae, respectively. The fact that the basal Crassula clade was a sister to all other members of the Crassulaceae family implies that it plays a key role in the evolution of the family. Of particular note, our phylogeny found that the Kalanchoideae was closely related to the Sempervivoideae, suggesting these two groups share a relatively recent common ancestry. This finding agrees with previous studies that proposed a close relationship between these two subfamilies.

The Sempervivum clade, consisting of the sections Jovibarba and Sempervivum, forms a monophyletic genus that is positioned as a sister group to the Aeonium, Leucosedum, and Acre clades within the Sempervivoideae subfamily. Nevertheless, the status of the Jovibarba section has remained uncertain for many years [6,10,50]. Both Jovibarba and Sempervivum share similarities, such as the base chromosome number [58] and phytochemistry [11], but differ in the flower petal count (6 vs. 8–16), presence of stolons (absent vs. present) [59], and pollen grain size [12]. Some researchers advocated for merging the genus *Jovibarba* into *Sempervivum* [6,50]. On other hand, molecular evidence, including ITS and IGS data, show that Jovibarba and Sempervivum are two distinct genera [7]. In this study, the robust monophyly of the Jovibarba section suggests that it represents a distinct evolutionary lineage within this genus. Whether these two represent separate genera or not requires further investigation with a broader sampling of species.

## 5. Conclusions

Our study provides eight newly assembled plastomes of the *Sempervivum* species. The comparative genomics of the *Sempervivum* plastomes show that their genomes were conserved in genome size (150,845–151,473 bp) and GC content (37.57–37.63%). Then, 14 HVRs (π value: 0.01173–0.02702) were found in *Sempervivum* plastomes. And in the IR junction pattern, the *ycf1* and *rps19* genes were found to span both the SSC and IRa regions. In the CUB analysis, both the ENC plots and neutrality plots suggest that the CUB of the genus *Sempervivum* plastid genes was predominantly governed by natural selection. Finally, our phylogenetic studies confirmed the monophyletic status of the genus *Sempervivum* and further resolved the monophyletic clade into two distinct sections: Jovibarba and Sempervivum.

## Figures and Tables

**Figure 1 genes-15-00441-f001:**
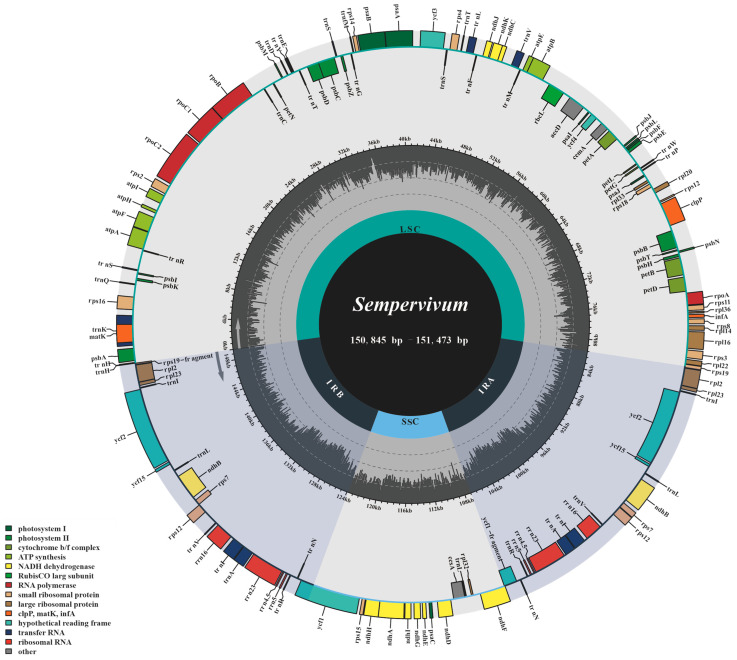
Plastome annotation map within eight *Sempervivum* taxa. Directed with arrows, genes that are listed inside and outside of the circle are transcribed clockwise and counterclockwise, respectively. Genes are color-coded by their functional classification.

**Figure 2 genes-15-00441-f002:**
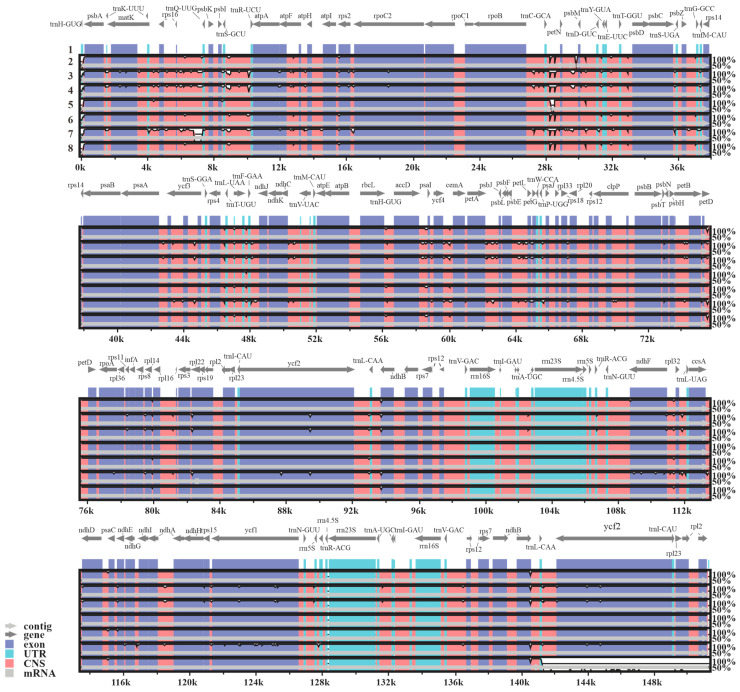
Structural comparisons of eight newly sequenced *Sempervivum* plastomes with the mVISTA program. The Y-axis represents the consistency of sequences, where the numbers ranged from 50% to 100%, and the X-axis means the plastomic gene position. The labels 1 to 8 represent *S. arachnoideum*, *S. calcareum*, *S. globiferum*, *S. globiferum* subsp. *hirtum*, *S. funckii*, *S. ciliosum*, *S. heuffelii*, and *S. tectorum*, respectively.

**Figure 3 genes-15-00441-f003:**
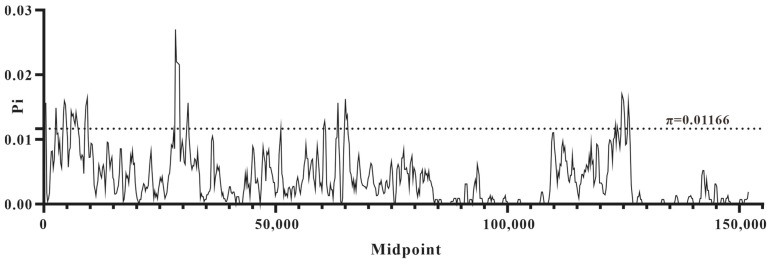
Sliding window analyzes of DNA polymorphisms of eight *Sempervivum* plastomes (window length: 600 bp; step size: 200 bp). The X-axis represents the midpoint position of each window, and the Y-axis shows the π value of each window. Regions with π > 0.01166 were identified as HVRs.

**Figure 4 genes-15-00441-f004:**
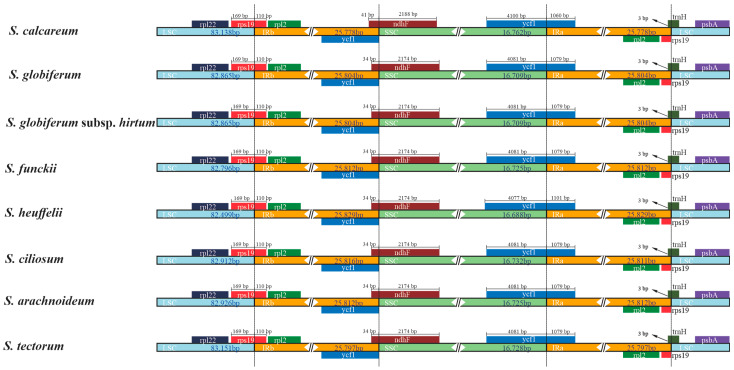
Comparison of the LSC, SSC, and IR region borders of 8 *Sempervivum* plastomes. The LSC, IR, and SSC regions are colored with blue, orange, and green blocks, respectively. Gene boxes above the block were transcribed counterclockwise, while those below the block were transcribed clockwise.

**Figure 5 genes-15-00441-f005:**
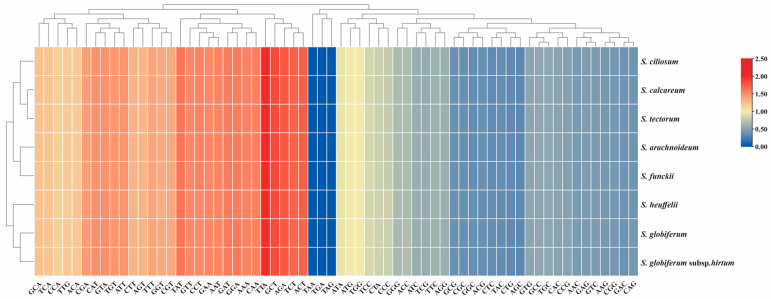
The heatmap of overall RSCU values among *Sempervivum* species based on 53 CDSs (length ≥ 300 bp). The x-axis represents the cluster of codons, and the y-axis represents the clusters of species.

**Figure 6 genes-15-00441-f006:**
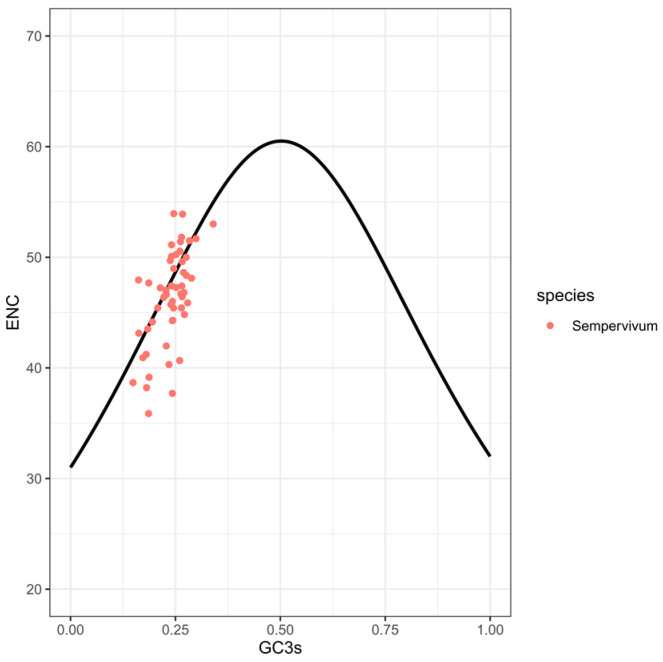
The ENC plot for 53 CDSs from *Sempervivum*, with GC3s as the x-axis and ENC as the y-axis. And the expected ENC values (standard curve) were calculated according to following formula: ENC = 2 + GC3s + 29/[GC3s^2^ + (1 − GC3s)^2^] [39].

**Figure 7 genes-15-00441-f007:**
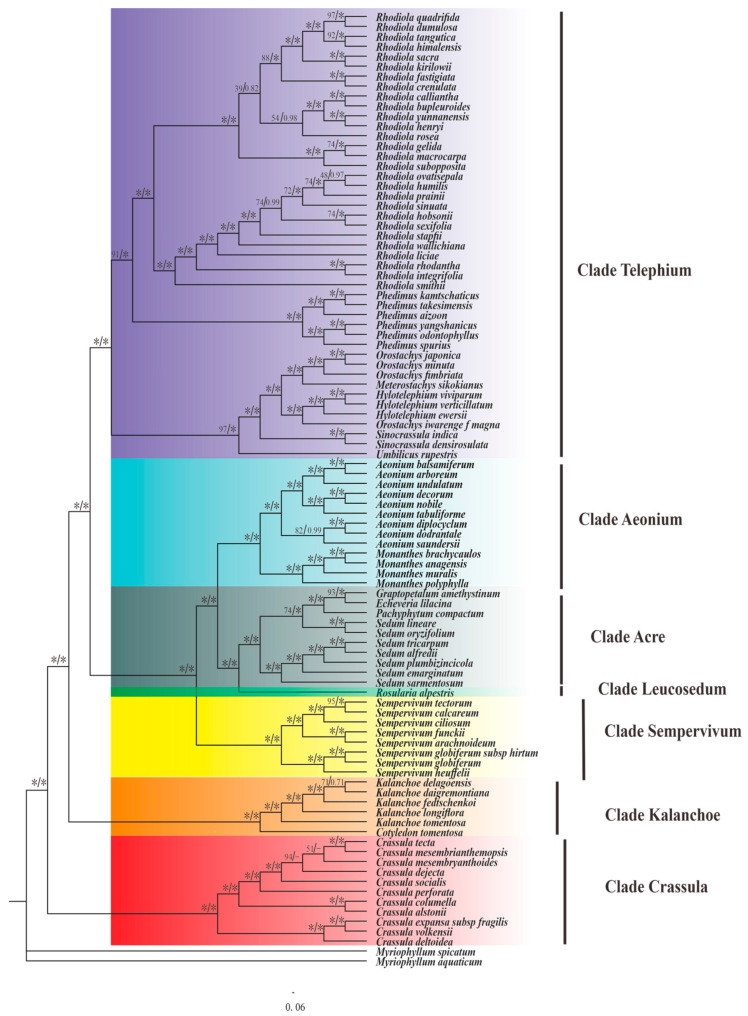
The phylogenetic tree from 94 Crassulaceae species found using maximum likelihood (ML) and Bayesian inference (BI) methods. The maximum likelihood bootstrap (BS) values and Bayesian posterior probability (PP) values are displayed at each node; * denotes nodes with a bootstrap value of 100% or a posterior probability of 1.00; - represents clades collapsed in the strict consensus tree.

**Table 1 genes-15-00441-t001:** Analyses of optimal codons in the plastomes of eight *Sempervivum* taxa.

Taxa	Codon	AA	High	Low	ΔRSCU
*S. tectorum*	CTA	Leu	1.43	0.80	0.63
	CGT	Arg	1.79	0.97	0.81
	GGG	Gly	1.05	0.63	0.42
*S. heuffelii*	CTA	Leu	1.43	0.92	0.52
	CGT	Arg	1.79	0.93	0.85
*S. ciliosum*	CTA	Leu	1.43	0.80	0.63
	CGT	Arg	1.79	0.97	0.81
*S. funckii*	CTA	Leu	1.43	0.80	0.63
	CGT	Arg	1.79	0.97	0.81
*S. globiferum* subsp. *hirtum*	CTA	Leu	1.43	0.80	0.63
	CGT	Arg	1.79	0.93	0.86
*S. globiferum*	CTA	Leu	1.43	0.80	0.63
	CGT	Arg	1.79	0.93	0.86
*S. calcareum*	CTA	Leu	1.43	0.80	0.63
	CGT	Arg	1.79	0.97	0.81
	GGG	Gly	1.05	0.63	0.42
*S. arachnoideum*	CTA	Leu	1.43	0.80	0.63
	CGT	Arg	1.79	0.97	0.81

## Data Availability

The eight plastome data sequences generated in this study are available in GenBank of the National Center for Biotechnology Information (NCBI) (https://www.ncbi.nl-m.nih.gov/nuccore (accessed on 28 March 2024)) under the access numbers: PP262169–PP262176.

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
