# Peer review of "Exploring Plastomic Resources in Sempervivum (Crassulaceae): Implications for Phylogenetics"

_genes, 2024, doi:10.3390/genes15040441_

Round 1

Reviewer 1 Report

Comments and Suggestions for Authors

In the manuscript "Exploring Plastomic Resources in Sempervivum (Crassulaceae): Implications for Phylogenetics" the authors present a study that is scientifically sound and well structured, with a focus on the codon usage and phylogeny of the Crassulaceae family.

I do have a few points to improve and discuss.

"The protein-coding genes (PCGs) were manually modified through BLAST searches [30]." What does this mean?

MAFFT7? Version 7: The algorithm is MAFFT.

"600 bp and a step size of 200 bp,"  what was the criteria for choosing these windows and steps?

In Figure 1, the circle plot labels can hardly be seen.

In the method, it is stated that "it did not contain any intermediate stop codons," but then in the results, "four pseudo genes," how has this been identified as pseudogenes? And what kind of pseudogenes? non-processed, processed, or retrocopies?

Table S1 should not be in lower case

Table 1 is not cited in the main text, and when a table or figure is first presented in the text, it should include the main conclusion of the figure, table, or else.

"After analyzing the neutrality graph," these are plots, not graphs. Also, throughout the manuscript, it is often used to use plots and graphs as synagogues, which is not true. Also refer as often as needed; the reader should be guided by a figure or plot. This should be "After analyzing the neutrality plot (Figure S1)."

Also, all the data for the phylogentic tree should be shared, including the alignments and newick tree. This is essential for the reproducibility of the results and open science. 

Reviewer 2 Report

Comments and Suggestions for Authors

Here’s my review on the manuscript entitled as “Kan et al., 2024, Exploring Plastomic Resources in Sempervivum (Crassulaceae): 2 Implications for Phylogenetics” focusing on sequence analyses of plastomes of seven species of Sempervivum (Crassulaceae) to examine plastome organization, structural diversity, codon usage patterns, and phylogenetic relationships within the genus. The authors made different analyses on structure comparison, codon usage and aversion patterns, and phylogenetic relationship. The purpose the study was to fill the gap by generating relevant data that may help for further comparative genomics as well as to understand the evolution of Sempervivum with possible molecular markers for DNA barcoding. Theie indicated that codon usage patterns in the plastomes caused by selective pressures (natural selection) and CUB could serve as a valuable tool for inferring evolutionary relationships among species.

The manuscript is well articulated and relevant data were generated with sound interpretation. I would like to forward my concerns that should be addressed for consideration of this manuscript.

1. In the introduction parts of the manuscript, the economic use or values of Crassulaceae family are not yet mentioned with worldwide research activities. I would like to suggest that such an important point should be included as a literature review in the introduction part of the manuscript.

2. What are the authors basic criterion and reason (distinct features) morphological and genetic differences for the selection of eight houseleek (Sempervivum) taxa such as S. globiferum, S. globiferum subsp. hirtum, S. heuffelii, S. calcareum, S. funckii, S. ciliosum, S. arachnoideum and S. tectorum as representative samples for this study? This issue should be briefed on in materials and methods with justification about each spp. representativeness.

3. Is it standardized approach for consideration of PCGs with the top 5% highest and lowest ENC values with ΔRSCU values greater than 0.08 and RSCU values meeting the conditions of low ENC group > 1, and high ENC group < 1? What is your confidence level that can be justified statistically?

4. What is the contribution of plastomes genetic materials in relation to morphological phenotypic characterization in terms of applied research beyond taxonomical classification values?

5. Typo error that should be corrected

a. “rps8 gene” line# 212 should be italic.

b. “O these clades” line# 283

c. And others
